# Immunogenicity and inflammatory properties of respiratory syncytial virus attachment G protein in cotton rats

Margaret E. Martinez[1]*, Cristina Capella Gonzalez[2], Devra Huey[1], Mark E. Peeples[3,4], Douglas McCarty[5], Stefan Niewiesk[1]

1 Department of Veterinary Biosciences, The Ohio State University College of Veterinary Medicine, Columbus, Ohio, United States of America, 2 Biomedical Research Center Battelle Memorial Institute, Columbus, Ohio, United States of America, 3 Center for Vaccines and Immunity, Abigail Wexner Research Institute at Nationwide Children's Hospital, Columbus, Ohio, United States of America, 4 Department of Pediatrics, The Ohio State University College of Medicine, Columbus, Ohio, United States of America, 5 Rare Disease Research Unit, Pfizer, Morrisville, North Carolina, United States of America

* martinez.610@osu.edu

**Data Availability Statement:** All relevant data are within the manuscript and its Supporting Information files.

## Abstract

Human respiratory syncytial virus (RSV) is a leading cause of lower respiratory tract infection in infants and young children worldwide. The attachment (G) protein of RSV is synthesized by infected cells in both a membrane bound (mG) and secreted form (sG) and uses a CX3C motif for binding to its cellular receptor. Cell culture and mouse studies suggest that the G protein mimics the cytokine CX3CL1 by binding to CX3CR1 on immune cells, which is thought to cause increased pulmonary inflammation in vivo. However, because these studies have used RSV lacking its G protein gene or blockade of the G protein with a G protein specific monoclonal antibody, the observed reduction in inflammation may be due to reduced virus replication and spread, and not to a direct role for G protein as a viral chemokine. In order to more directly determine the influence of the soluble and the membrane-bound forms of G protein on the immune system independent of its attachment function for the virion, we expressed the G protein in cotton rat lungs using adeno-associated virus (AAV), a vector system which does not itself induce inflammation. We found no increase in pulmonary inflammation as determined by histology and bronchoalveolar lavage after inoculation of AAVs expressing the membrane bound G protein, the secreted G protein or the complete G protein gene which expresses both forms. The long-term low-level expression of AAV-G did, however, result in the induction of non-neutralizing antibodies, CD8 T cells and partial protection from challenge with RSV. Complete protection was accomplished through co-immunization with AAV-G and an AAV expressing cotton rat interferon α.

## Introduction

Human respiratory syncytial virus (RSV) is the leading cause of lower respiratory disease in infants and children worldwide. RSV has a global burden of hospitalization of 19.2 per 1,000

**Funding:** M.E.M. was funded through the Genetech Fellowship under grant number GRT00044407 from Genentech (https://urldefense.com/v3/__ https://www.gene.com/__;!!KGKeukY!kK079 YjJDBxIYb4cCd4SEhlt14tL5fghqOewU8tfu KWiRtOTljnrsH7ujR1LFKsMRVtl$). D.H. and S.N. were funded through grant number P01AI112524 from the National Institute of Health National Institute of Allergy and Infectious Diseases (https:// urldefense.com/v3/__https://www.niaid.nih.gov/__ ;!!KGKeukY!kK079YjJDBxIYb4cCd4SEhlt14tL5f ghqOewU8tfuKWiRtOTljnrsH7ujR1LFHaJx-S8$). M.P and C.C-P. were funded through grant number P01 AI112524 and R01 AI093848 rom the National Institute of Health National Institute of Allergy and Infectious Diseases (https://urldefense.com/v3/__ https://www.niaid.nih.gov/__;!!KGKeukY!kK079 YjJDBxIYb4cCd4SEhlt14tL5fghqOewU8 tfuKWiRtOTljnrsH7ujR1LFHaJx-S8$). Genentech provided the corresponding author with a research fellowship that aided in the following research to be conducted. Dr. Douglas McCarty developed and provided the adeno-associated virus vectors used for the experiments in the following manuscript, as well as edited the manuscript for publication. Though Dr. Douglas McCarty is now employed by Pfizer, he did not conduct studies nor contribute to the study plan while at Pfizer. Only editing of the manuscript was performed while he was employed by Pfizer. Therefore, Pfizer did not contribute funding towards study design, materials, data collection or the majority of the salary for Dr. Douglas McCarty for the research presented in the following manuscript. The funder provided support in the form of salaries for authors [DM], but did not have any additional role in the study design, data collection and analysis, decision to publish, or preparation of the manuscript. The specific roles of these authors are articulated in the 'author contributions' section.

**Competing interests:** Genentech provided the corresponding author with a research fellowship that aided in the following research to be conducted. One of the authors was employed by Pfizer partly during the completion of the present manuscript. This does not alter our adherence to PLOS ONE policies on sharing data and materials.

children and a burden of 6.6 deaths per 1,000 children <1 year of age [1]. Currently there is no vaccine or therapeutic available to prevent or treat infection. The only preventive measure for RSV infection is the injection of a monoclonal antibody that is available for at risk infants. An early attempt to develop an RSV vaccine by using formalin-inactivated RSV failed and actually led to high levels of inflammation and more severe clinical disease after natural exposure [2–4]. In the development of RSV vaccines, there is debate as to whether the RSV attachment glycoprotein (G) should be incorporated into the subunit or modified live vaccines. Some argue its presence could cause a pro-inflammatory response possibly similar to that seen after immunization with the formalin-inactivated vaccine, while others argue its ability to induce neutralizing antibodies is beneficial for protection [5–7].

The G protein, one of three viral surface glycoproteins, is a type 2 membrane protein with an N-terminal membrane anchor and a 36 amino acid cytoplasmic domain (Fig 2). Its central region contains a completely conserved 13 amino acid sequence that includes part of a cysteine noose whose neck is formed by two disulfide bonds. The third and fourth of these cysteines compose a CX3C motif, two cysteines separated by 3 amino acids [8]. Flanking the central conserved region are two heavily glycosylated mucin-like regions [8]. The receptor-binding site contains a CX3C motif and is located between the two cysteines at amino acid positions 182 and 186. The CX3C motif of the G protein binds to the CX3C receptor 1 (CX3CR1), which is the host receptor for viral entry on human ciliated airway epithelium [9–11].

The G protein has two forms, a membrane bound form and a secreted form. The transmembrane domain, amino acids 36 to 67, includes a second translation start at codon position 48. Initiation at this site produces a secreted form of the G protein. Proteolysis near the N-terminal anchor domain releases the secreted form from the infected cell [12].

After natural infection, the G protein has been demonstrated to induce neutralizing antibodies in a minority of infected children. The presence of these G protein antibodies correlated with less severe disease [13]. In some animal studies, it has been demonstrated that G protein antibodies can protect against challenge with RSV [6,14–17]. However, other animal studies have pointed to the G protein as a cause of an adverse hypersensitivity-like inflammatory response similar to the one that occurred after immunization with the formalin-inactivated RSV vaccine [5,7,18–20]. The potential pro-inflammatory action of the G protein is based on its similarity to the chemokine CX3C ligand 1 (CX3CL1 (or fraktalkine)) which has a similar structure. As in the G protein, CX3CL1 has a CX3C motif that is involved in forming two disulfide bonds [21]. CX3CL1 binds through the CX3C motif to the CX3C receptor 1 (CX3CR1) on macrophages, monocytes, dendritic cells, B and T lymphocytes, natural killer cells, and epithelial cells [22,23]. Also similar to the RSV G protein, CX3CL1 is produced as a membrane bound and a secreted form. The membrane bound form is expressed on activated endothelium or epithelial cells and binds to CX3CR1 on inflammatory cells [24,25]. After cleavage by a metalloproteinase, the ectodomain is released as the secreted form of CX3CL1, which acts as a chemoattractant for those inflammatory cells that express CX3CR1 [24,26].

Many of the studies which address the pro-inflammatory properties of the G protein compared the inflammatory response of wild type RSV to an RSV with a deletion of the G protein gene or after neutralization with a G protein specific antibody and were compared to infection with a complete RSV [27–31]. These studies eliminated or decreased receptor binding capabilities of RSV via the G protein (which was not known at the time) thus making it impossible to differentiate the impact of a lack of viral replication on inflammation from the pro-inflammatory capacity of the G protein. In order to separate the receptor binding function of the G protein from any chemokine function, we used soluble G protein and expressed the G protein in an adeno-associated virus vector either as the membrane bound and/or secreted form and investigated its inflammatory potential in a cotton rat model. Through this model we were

able to determine the effects of the G protein in its various forms on the immune response and its ability to protect against RSV challenge.

## Materials and methods

### Cotton rats

Inbred male and female cotton rats (Sigmodon hispidus) were purchased from Envigo (Indianapolis, Indiana) and housed as previously described [32,33] in polysulfone microisolation cages (NextGen Rat 900, Allentown Inc., Allentown, NJ, USA) in a barrier facility with a 12:12 hour light cycle. Cotton rats were maintained at $20 \pm 2°C$ and 30% to 70% relative humidity. Cotton rats were 4–8 weeks of age, and were euthanized by carbon dioxide inhalation. All studies were approved by the Institutional Animal Care and Use Committee of The Ohio State University.

### Inoculation and Infection with protein, virus and house dust mite (HDM) antigen

Animals were anesthetized as previously described [32] with isoflurane inhalation and intranasally inoculated with a fluid volume of 100μL. For bronchoalveolar lavage studies, cotton rats were inoculated with either 5μg of purified G protein, or 5μg or 20μg recombinant mouse CX3CL1 (458-MF, R&D Systems, Minneapolis, MN, USA). To determine the best serotype of AAV for studies in the cotton rat respiratory tract, cotton rats were inoculated with $10^{9-10}$ DNAase resistant particles (DRP) with different serotypes of AAV expressing GFP for flow cytometry studies. Blood samples were obtained by retro-orbital bleed in isofluorane narcosis. For the studies of inflammation and immunogenicity, cotton rats were inoculated with $2x10^{10}$ DRP of various AAV constructs and challenged with $10^5$ $TCID_{50}$ of RSV A2. 100μg house dust mite (HDM, *Dermatophagoides pteronyssinus*) antigen was absorbed to aluminum phosphate (AdjuPhos, Brenntag, Ballerup, Denmark) at a 1:1 ratio for 30 minutes at room temperature and injected into cotton rats intraperitoneally (IP). Sensitization was followed 8 days later with intranasal administration of 100μg HDM in a volume of 100μL PBS. Four days after HDM administration, cotton rats were euthanized through $CO_2$ inhalation and lungs were collected for histologic examination.

### G protein ectodomain purification

The transfection and purification of the ectodomain of the RSV G protein was performed as previously published [13]. The cytoplasmic and transmembrane domains were removed from the codon optimized G protein gene and replaced with the N-terminal region of the Schwarz measles virus H protein followed by a furin cleavage site. The G protein gene was then inserted into the pcDNA3.1 plasmid. Freestyle 293-F cells were transfected with the plasmid expressing the G protein using a cationic lipid-based formulation. The G protein was captured using agarose-bound wheat germ agglutinin (WGA) (Vector Laboratories), washed with 0.15 M NaCl, 10 mM HEPES (pH 7.5), and eluted with chitin hydrolysate (1:10 dilution) in 0.15 M NaCl, 10 mM HEPES (pH 3.0). The eluate was neutralizing with 10mM HEPES (pH 7.5), concentrated and purified by size-exclusion chromatography on a HiLoad Superdex 200pg gel filtration column (GE Healthcare). Protein concentrations were determined by using the Nanodrop 2000c (Thermo Fisher) and Pierce BCA protein assay (Thermo Fisher). G protein purity was determined by displaying proteins on polyacrylamide gels and staining with the Glycoprotein Staining Kit (Pierce). Protein purity was estimated to be >95%.

## RSV production and titration

Stocks of RSV A2 were grown in human epithelial (HEp2) cells as described [32,33]. HEp2 cells were infected with an MOI of 0.001 of RSV A2 in MEM for 1 hour at 37˚C and incubated in 30 mL MEM/2% fetal calf serum at 37˚C for 4 days. Once infection yielded a cytopathic effect of approximately 80%, 3mL of 1M MgSO4/0.25M HEPES was added and cells were scraped from the flask with a cell scraper. The cells and medium were briefly frozen at –80˚C, thawed, and centrifuged at 3000rpm for 15 min at 4˚C. The supernatant was collected and centrifuged through a 15 mL 35% sucrose cushion at 15,000 rpm in an SS 34 rotor (Sorvall, Thermo Fisher Scientific, Waltham, MA) for 5 h at 4˚C. Virus pellets were resuspended in MEM containing 10% trehalose.

Viral stocks were titrated with the tissue culture infectious dose 50 assay. The same assay was used for virus determination from virus infected noses and lungs. Nasal turbinates and left lung lobes were collected from euthanized cotton rats. Tissues were weighed and homogenized in 3mL MEM. Lungs were homogenized using a Precellys 24 tissue homogenizer (Bertin Instruments, Montigny-le-Bretonneux, FRANCE,) following the manufacturer's recommendations. The mucosa of the nasal turbinate was homogenized with mortar and pestle with the addition of sterile sand. Ten-fold serial dilutions of the homogenates (100 μL/well) were added to a 48-well plate with 80 to 90% confluent HEp-2 cells. After one hour, the monolayer was washed three times with MEM, and 500uL of MEM/2% FCS was added per well. After five days, plates were scored microscopically for cytopathic effect. The amount of virus inocula was expressed as the quantity of virus that could infect 50% of inoculated tissue culture monolayers ($TCID_{50}$). $TCID_{50}$ was calculated according Reed and Muench [34,35].

## AAV viral vector construction and production

The AAV vector plasmid, pHpa-trs-KS [36], is used to generate the recombinant self-complimentary AAV vector under the control of a cytomegalovirus (CMV) promoter with an SV40-derived mini intron and containing two sequential polyadenylation signals (SV40PA) as previously described [37]. Gene inserts were inserted at the gene junction between the SV40 intron segment and the SV40PA segment using the restriction sites Age 1 and Not 1. The gene inserts and AAV vector plasmid were digested with Age 1 and Not 1 enzymes (NEB) at 37˚C for 6–8 hours and purified using GeneJet PCR Purification kit (K0701, ThermoFisher Scientific, Waltham, MA, USA) according to manufacturer's recommendations. Gel electrophoresis and extraction was performed on the digested and purified transgenes and AAV vector plasmid using a 1% agarose gel and QIAquick Gel Extraction Kit (28506, Qiagen, Germantown, MD, USA). The inserts were ligated into the AAV vector plasmid using T4 DNA Ligase according to the manufacturer's recommendations (ThermoFisher Scientific, Waltham, MA, USA). One Shot TOP10 bacteria (C404010, ThermoFisher Scientific, Waltham, MA, USA) were transformed and plated on agar plates with LB broth and 100μg/mL ampicillin. Single colonies were expanded and plasmids were purified with a Qiagen Plasmid Kit (Qiagen, Germantown, MD, USA).

**G protein gene inserts.** The gene sequence for the codon optimized RSV A2 G protein sequence has been published [38]. Based on this gene sequence the nucleotide at position 144 was changed from a guanine to a cytosine which resulted in a change of the second start codon at amino acid position 48 to an isoleucine to produce the membrane bound only G protein. To develop the sequence for a G protein with a dysfunctional receptor binding site, the nucleotide at position 557 was mutated from a guanine to a cytosine which results in a change at amino acid position 186 from a fourth cysteine in the cysteine noose/receptor binding site to a serine (see S1 Fig).

These sequences were submitted to Integrated DNA Technologies (Coraville, IA, USA) for gBlock synthesis of double stranded DNA fragments. The G protein genes were amplified using high fidelity Taq polymerase, a G protein forward primer with an Age I restriction site (5' agactaag accggt atgagcaagaacaaggaccagcgga 3') or a sG protein forward primer with an Age I restriction site (5' agactaag accggt atgatcatcagcaccagcct gatca 3'), and a G protein reverse primer with a Not I digestion site (5' cttagtct gcggc cgc ctactgccgaggggtgttgggagggctgct 3') at a denaturing temperature of 94˚C, an annealing temperature of 60˚C and an extension temperature of 74˚C for 35 cycles. The sequence of the amplified DNA fragments was confirmed by gel electrophoresis and Sanger sequencing. Gene inserts were digested with the respective enzymes and inserted by ligation into the AAV plasmid.

**Cotton rat interferon alpha insert.** Naïve cotton rat splenocytes were isolated through a 100nm sieve and plated in duplicate at a concentration of $10^6$cells/well in a sterile 12 well plate. Splenocytes were incubated at 37˚C in RPMI/10% FCS and 10μg/mL TLR9 agonist (ODN2216, InvivoGen, San Diego, CA, USA) overnight. Total RNA was extracted and purified from stimulated splenocytes with the RNeasy Mini Kit (74106, Qiagen, Germantown, MD, USA) according to the manufacturer's recommendations. Complimentary DNA was transcribed from the purified total RNA using SuperScript III First-Strand Synthesis System (18080051, ThermoFisher Scientific, Waltham, MA, USA) according to the manufacturer's recommendations using cotton rat interferon alpha specific primers based on the published sequence [39] with Age I and Not I restriction sites (Forward 5' agactaag accggt atgtccaggtcatgtgctttc; Reverse 5' cttagtct gcggccgc ttacttcttctcctcactccatct). The cotton rat interferon alpha gene was amplified from cDNA using high fidelity Taq polymerase, using the same forward and reverse primers with an Age 1 and Not 1 restriction site respectively at a denaturing temperature of 94˚C, an annealing temperature of 55˚C and an extension temperature of 74˚C for 35 cycles. The amplified sequence of the cotton rat interferon alpha DNA fragment was confirmed by gel electrophoresis and Sanger sequencing, and subsequently used as insert for the AAV plasmid (described above).

**Adeno associated virus vector production.** AAV plasmids containing different versions of the G protein gene were submitted to the Viral Vector Core of The Research Institute at Nationwide Children's Hospital (Columbus, Ohio, USA) for AAV production as described [36]. Briefly, recombinant self-complementary AAV vector plasmids containing the genes of interest were transfected into adherent HEK293 cells with AAV serotype 5 and Ad helper plasmids by $CaPO_4$ transfection. Cell were fragmented and AAV particles were purified by iodixanol step gradient ultracentrifugation and ion exchange column chromatography [40,41]. AAV titers were determined by qPCR and purity was confirmed by SDS-PAGE as previously described [40,41].

## Broncho-alveolar lavage

Broncho-alveolar lavage fluid (BALF) was collected as previously described [32]. Briefly, BALF was collected from cotton rats immediately after euthanasia through $CO_2$ inhalation. The trachea was cannulated, and the lungs were lavaged with 1mL of PBS supplemented with 1% protease-free BSA. The BALF was kept on ice until processed by the Comparative Pathology and Mouse Phenotyping Shared Resource, Ohio State University. Automated nucleated cell counts were performed by using a Forcyte veterinary hematology analyzer (Oxford Science, Oxford, United Kingdom).

## Histology and immunohistochemistry

After euthanasia, the lungs were inflated with 1mL of 4% paraformaldehyde (PFA) and fixed in 4% PFA. All tissues were routinely processed, embedded in paraffin wax, and sectioned

(4μm) according to the revised guide for organ sampling and trimming in rats and mice by the European Experimental and Toxicologic Pathology group [42]. Cotton rat lung histology slides were routinely stained with hematoxylin and eosin (H&E). Light microscopic evaluation was performed by a veterinary anatomic pathologist (MM) (model CX41, Olympus, B and B Microscopes Limited, Pittsburgh, PA, USA). A semi-quantitative scoring system, adapted from previous studies [43], was established and applied (S2 Table). The sum of the mean scores were calculated to produce a total histologic inflammatory score. Immunohistochemistry was performed on the formalin fixed paraffin embedded slides using goat anti-RSV polyclonal antiserum (Virostat, cat# 0601, Westbrook, Maine, USA) at a dilution of 1:800. A semi-quantitative immunohistochemical scoring system was devised (S1 Table) to compare the expression of the RSV G protein based on percentage of cells staining as well as strength of stain. The sum of the mean scores was calculated to produce a total immunohistochemical score.

## Flow cytometry

Cotton rats were inoculated intranasally with $10^{10}$ DRP of AAV serotypes 1, 5, and 6 expressing GFP [37,44]. Five days post-infection, tracheas were digested with HBSS without Ca2+ or Mg2/1 mM EDTA (HBSS/EDTA)/0.2 mg/mL of Collagenase for 15 minutes. 1–5 x $10^5$ of the isolated epithelial cells were washed with RPMI + 10%FCS and analyzed by flow cytometry for expression of GFP (FACS Calibur flow cytometer (BD Biosciences). Data were analyzed with FlowJo v.10 software (Tree Star, Inc., Ashland, OR).

## RSV specific IgG ELISA

For enzyme-linked immunosorbent assay (ELISA), a plate was coated with 5 μg/ml of gradient-purified, UV-inactivated RSV in 200 mM $NaCO_3$ buffer (pH 9.6) at a total volume of 50–100μL at 4°C overnight, blocked with phosphate-buffered saline (PBS)/10% FCS/0.05% Tween 20 for 2 hours at room temperature. To test for RSV-specific cotton rat IgG, RSV-coated plates were incubated with dilutions of cotton rat serum at 4°C. After being washed, plates were incubated with chicken anti-cotton rat IgG-HRP (PAB29753, Abnova Corporation, Taipei City, Taiwan, 1:1000 dilution) for 2 hours. Plates were washed and developed with TMB (3, 3', 5, 5'–tetramethylbenzidine) substrate (50 μL/well). Color development was stopped with 1M phosphoric acid. Optical density values were determined at 450nm (OD450).

## RSV neutralization assay

Cotton rat serum samples were two-fold serially diluted and incubated with 50 $TCID_{50}$/well RSV A2 for one hour at 37°C in a 96-well flat-bottom plate. One hour post-incubation $10^4$ HEp-2 cells were added per well. Five days post-infection cytopathic effect (CPE) was determined microscopically. The titer was defined as the reciprocal of the last protective serum dilution, as calculated from duplicate measurements.

## CD8+ T cell depletion

Cotton rats were inoculated on day −1, 1 and 3 after RSV infection with 0.5mg of a cotton rat CD8 alpha specific monoclonal antibody CD8[+] [33,45] obtained from Sigmovir Inc., Rockville.

## Statistics

Statistics were performed using GraphPad Prism Software version 6.07 (San Diego, CA, USA). Differences between groups were analyzed using a One-Way ANOVA or unpaired two-tailed

student T test, with Tukey's multiple comparison posttests. P < 0.05 was considered to be statistically significant. All data are represented as means +/- standard deviations with individual data points. The total number of animals per experimental group is indicated in the figure legends.

## Results

### CX3CL1 or the G protein ectodomain do not induce pulmonary inflammation

CX3CL1 is upregulated in the course of various inflammatory diseases as well as in lungs of infants after RSV infection [46–48]. In mice, CX3CL1 has been shown to attract leukocytes to sites of injury in the brain and blood vessels [22,24,49]. Because the G protein binds to the same receptor as the endogenous chemokine CX3CL1, it has been hypothesized that the G protein would also induce pulmonary inflammation through chemokine mimicry [11,31,50]. *In vitro*, CX3CL1 and the RSV G protein were reported to have similar levels of chemotactic activity on murine splenocytes [31]. Other *in vitro* studies using human cells also found a chemotactic property of the G protein, which depended on the CX3C motif [30,50]. However, there is little information about the *in vivo* chemotactic properties of the G protein. Therefore, we intranasally inoculated cotton rats with 5μg of purified G protein or CX3CL1. The dosage was selected based on previous *in vitro* studies as well as to recapitulate the relative physiologic amount of protein produced during natural infection [50,51]. We also selected the complete ectodomain of CX3CL1 versus studies that used solely the ectodomain to best compare to the G protein ectodomain, which also has mucin like domains, and to reflect the entire physiologic state of the chemoattractant form of CX3CL1. The total white blood cell count (WBC) in bronchoalveolar lavage fluid (BALF) was compared at 12, 24, and 48 hours post-inoculation to determine the optimal time point of peak *in vivo* chemotaxis, which was found to be 48 hours (S2 Fig). Forty-eight hours post-inoculation with PBS, cotton rats had a mean WBC of 349 +/- 79 per μL, cotton rats inoculated with 5 μg of CX3CL1 had 478 +/- 62 WBC/μL, those inoculated with G protein had a mean WBC of 404 +/- 96 per μL, and those inoculated with $10^5$ TCID$_{50}$ RSV had 421 +/- 133 WBC/μL. There was no statistically significant difference in total WBC counts in BALF among these groups (Fig 1). However, in this system even a superphysiological dose (20μg) of the natural ligand CX3CL1 did not result in a significant increase in WBC in BALF (S3 Fig) and therefore we explored alternate methods of G protein application.

### G protein mutants construction and expression within the adeno-associated virus vector

One reason for the lack of increase in pulmonary inflammatory infiltrates post-inoculation with the purified G protein may be that the protein was not expressed in the respiratory epithelium of cotton rats and also not in its natural form as the membrane-bound/secreted form. In order to address these issues, we utilized an adeno-associated virus vector system for long-term expression of the G protein in cotton rat bronchiolar epithelium. In order to define the AAV serotype best suited for cotton rat lung tissue, GFP expressing AAV vectors of serotypes 1, 5, and 6 were inoculated into cotton rats. Five days after inoculation, cotton rat tracheal epithelial cells were isolated and GFP expression was determined by flow cytometry. AAV-GFP serotype 5 had the highest level of GFP expression at around 20% (S3 Table).

The complete codon optimized G protein gene [38] was inserted into the AAV vector plasmid in order to generate both the membrane bound and secreted forms [6,11]. To determine the effects that the cysteine noose and CX3C motif may have on inflammation and

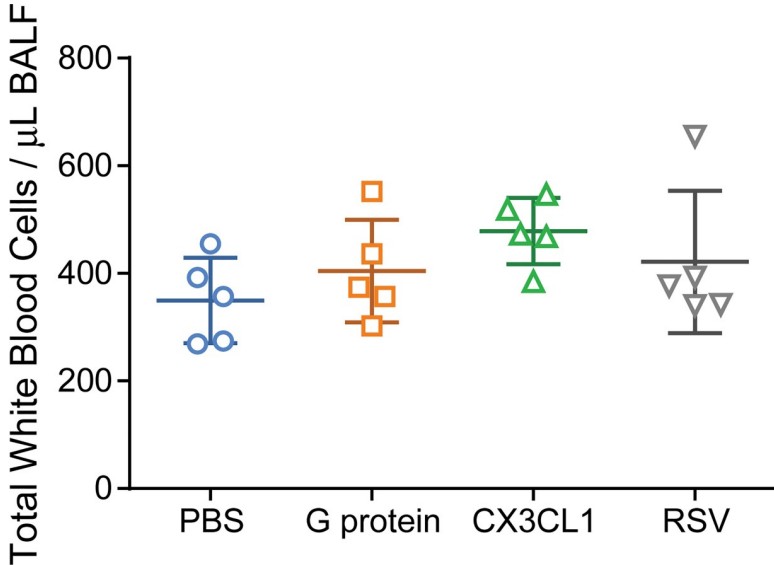

**Fig 1. Pulmonary inflammation 48 hours post-inoculation or infection.** The total number of white blood cells per μL of bronchoalveolar lavage fluid (BALF) was compared between cotton rats (n = 5) inoculated IN 2 days prior with 100μL of PBS, 50ug/mL purified G protein, 50ug/mL mouse CX3CL1, or $10^5$ TCID$_{50}$ RSV. There was no statistically significant difference among these groups; One-way ANOVA; p> 0.05. Data points represent mean total WBC/μL BALF for a single animal from a single experiment.

immunogenicity, we mutated the fourth cysteine in the noose at amino acid 186 to a serine (Fig 2B). In order to produce solely the membrane bound G protein, the second start codon was mutated from ATG to ATC (methionine to isoleucine; Fig 2C). To produce the secreted form of the G protein, the gene was inserted into the AAV vector plasmid beginning at the second start codon (Fig 2D). Correct insertion into the AAV vector plasmid as well as the correct sequence of the G protein variants was verified by sequencing.

Cotton rats were inoculated intranasally with $2x10^{10}$ DRP AAV serotype 5 expressing the G protein (AAV-G). After 5, 12, 19 and 26 days post-inoculation, lungs were harvested and analyzed by immunohistochemistry (IHC) for the expression of the G protein. G protein was detected in the cotton rat bronchiolar epithelium, in alveolar macrophages, and in type I pneumocytes by immunohistochemistry (Fig 3). At all time points, expression was most prominent within the type I pneumocytes and to a lesser degree in the bronchiolar epithelium. An immunohistological scoring system was devised (S1 Table) based on staining intensity and quantity. Expression was highest 5 days post-inoculation (mean IHC score 1.2 +/- 1) and was less than total protein expression after 5 days of RSV infection (mean IHC score 1.9 +/- 0.9, n = 45). However, the G protein could be detected up to 26 days post-inoculation (0.58 +/- 0.5) (S4 Fig). Immunohistochemistry demonstrated that not only is the G protein expressed *in vivo* and in the correct anatomical location, but that it is correctly formed in order for human anti-RSV antibodies to recognize. Also using the IHC scoring system, we were able to semi-quantitate that the amount of G protein expressed using the AAV vector system and compare it to total RSV protein expression.

## Long-term expression of the G protein via an AAV vector does not induce pulmonary inflammation

In cotton rats, RSV infection results in mild clinical disease with low pulmonary inflammation [32,52–57]. We tested whether long-term expression AAV-G within pulmonary epithelium

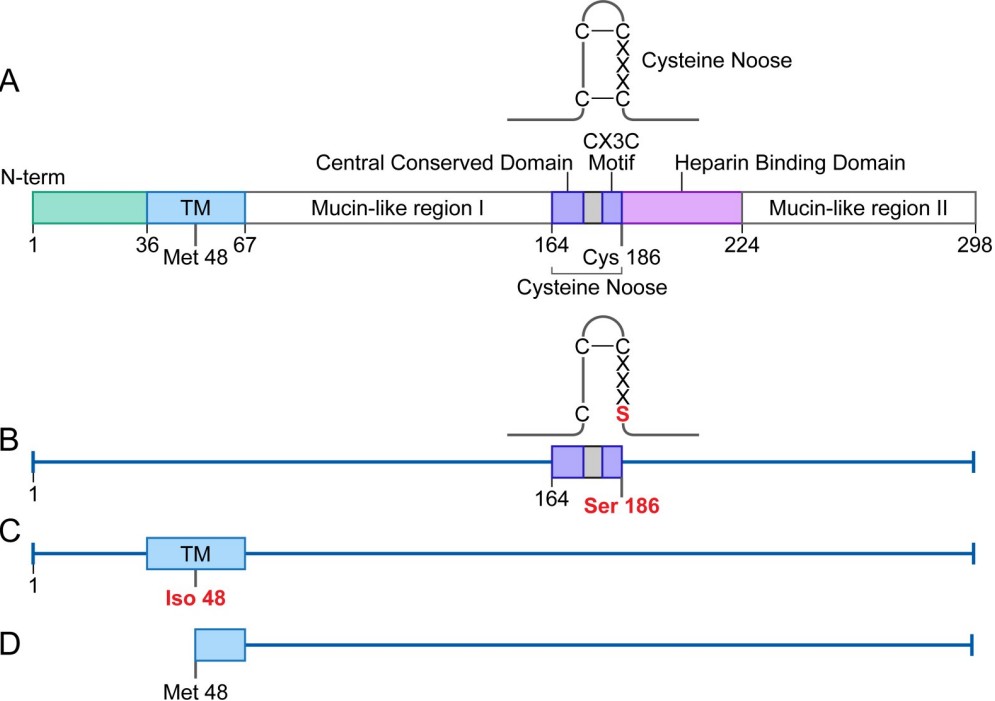

**Fig 2. Design of G protein mutations as inserted into the AAV vector constructs.** (A) Structure and gene of G protein with no alterations. The gene inserted into the AAV vector without any mutations produces the complete RSV G protein with both the membrane bound and secreted forms expressed and each with an intact receptor binding site (AAV-G). (B-D) The mutations to the G protein are indicated in the below figures. (B) AAV-G-C186S in which the fourth cysteine, amino acid 186, was mutated to a serine; and therefore, altered the CX3C motif/receptor binding site as well as disrupted the second disulfide bond within the cysteine noose. (C) AAV-mG is produced by mutating the G protein's second start codon, where the methionine was mutated to an isoleucine thus eliminating the secreted form's expression and only producing the membrane bound form. (D) The first start codon was deleted, so that protein translation of AAV-sG started at the second methionine at amino acid 48 resulting in the secreted form of the G protein.

would stimulate an inflammatory response, and compared it to RSV infection and a previously published allergy model (house dust mite (HDM) sensitization) which results in high pulmonary inflammation in cotton rats [32,55].

Inflammation was quantified based on the quantity of pulmonary inflammatory infiltrates (S2 Table). Histologic scores were compared in cotton rats inoculated 5, 12, 19, and 26 days after intranasal inoculation with AAV-G (Fig 4), with cotton rats infected for five days with RSV as this time point has been reported to demonstrate peak histologic inflammation [14], and with cotton rats inoculated with PBS or AAV-GFP. The highest histologic score (7.3 +/- 2.9) was determined on day 19 post-inoculation of AAV-G though it was not statistically different from the histologic scores at other time points or RSV infection. In comparison, the inflammatory positive control of house dust mite (HDM) sensitization had a significantly higher inflammatory score (13.2 +/- 1.2, One-way ANOVA, p<0.0001).

The assessment of inflammation by histologic scoring was complemented by the counting the number of of cells infiltrating the bronchoalveolar lavage fluid (BALF). As histological scoring demonstrated the highest level of inflammation associated with AAV-G protein was at day 19, we used this time point to compare animals inoculated with the same inoculum level as above ($2x10^{10}$ DRP) and with a 20-fold higher inoculum ($4.7x10^{11}$ DRP) of AAV-G. However, there was no difference in the influx of inflammatory cells as determined by BAL at either inoculum dose (Fig 5) in comparison to animals inoculated with AAV-GFP. Again, the

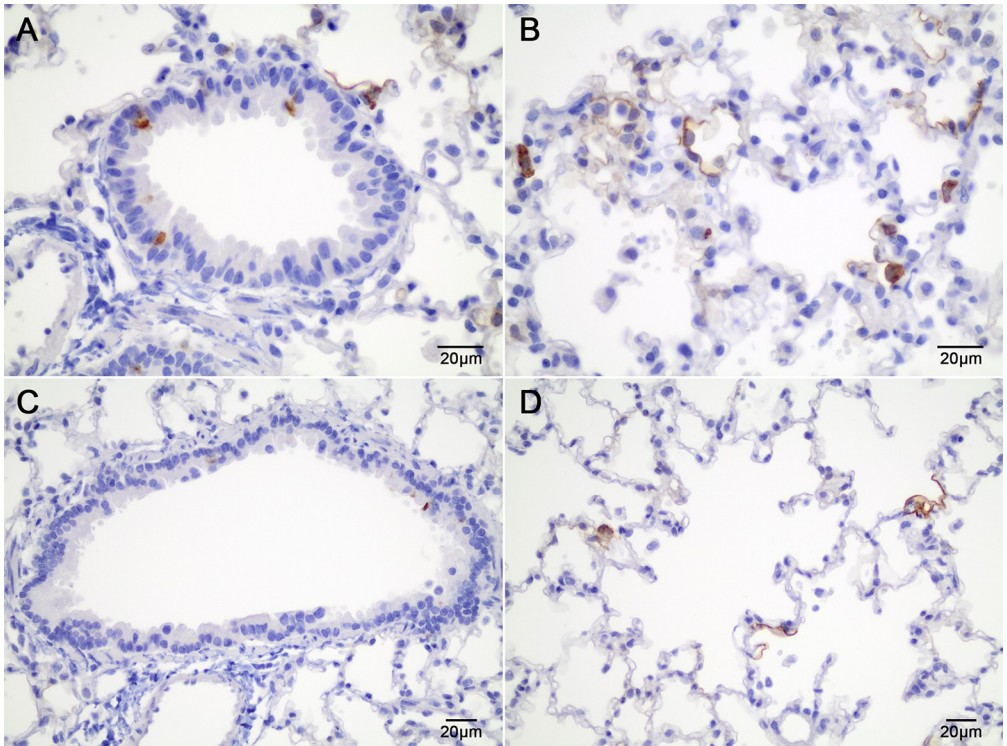

**Fig 3. AAV vector expression of the G protein in cotton rat lungs after intranasal inoculation.** (A) Immunohistochemical stain (IHC) to detect G protein expression using an anti-RSV antiserum in lungs from a cotton rat 5 days post intranasal inoculation with $2\times10^{10}$ DRP AAV-G. The dark brown cytoplasmic staining demonstrates G protein expression within the cotton rat bronchiolar epithelium. (B) IHC using an anti-RSV antiserum to detect RSV G protein 5 days post intranasal inoculation with $2\times10^{10}$ DRP AAV-G within alveoli. The dark brown cytoplasmic staining demonstrates G protein expression within type I pneumocytes and alveolar macrophages. (C) IHC using an anti-RSV antiserum to detect RSV G protein 26 days post intranasal inoculation with $2\times10^{10}$ DRP AAV-G within bronchioles. There is infrequent light brown cytoplasmic staining and thus fewer bronchiolar epithelial cells expressing the G protein. (D) IHC using an anti-RSV antiserum to detect RSV G protein 26 days post intranasal inoculation with $2\times10^{10}$ DRP AAV-G within alveoli. The light brown cytoplasmic staining demonstrates fewer type I pneumocytes and alveolar macrophages expressing the G protein.

inflammatory positive control of HDM sensitized cotton rats had significantly higher white blood cell counts in BALF compared to all other groups (One-way ANOVA, $p<0.0001$). In summary, our in vivo cotton rat experiments using inoculation of the purified G protein and the inoculation with AAV-G suggests that RSV G protein alone does not induce inflammation.

## The immunogenicity of the G protein and its protective qualities against RSV challenge

The G protein has been demonstrated to induce an antibody response in infants that is often low but none-the-less correlates with decreased disease severity, and in several animal model studies G protein specific monoclonal antibodies have been found to be protective against RSV challenge [6,13–16]. As we could not detect a significant inflammatory response against the G protein, we investigated whether the AAV-G protein could be immunogenic and protective.

We inoculated $2\times10^{10}$ DRP AAV-G intranasally into cotton rats and collected sera in weekly intervals. Immunized cotton rats were challenged 19 days later with RSV, and viral

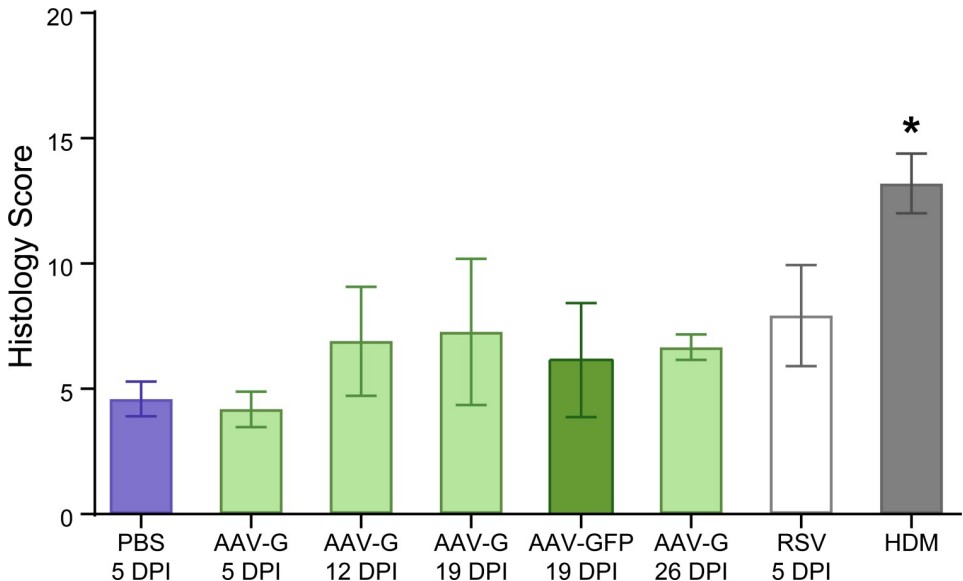

**Fig 4. Histologic assessment of inflammation in cotton rats inoculated with AAV-G compared to PBS, AAV-GFP, RSV-infected and HDM-sensitized cotton rats.** Semi-quantitative histologic scores representing total inflammatory cell infiltrates within cotton rat lungs inoculated IN with 100μL PBS 5 days post-inoculation (DPI), 5, 12, 19, and 26 DPI with $2 \times 10^{10}$ DRP AAV-G, 19DPI with $2 \times 10^{10}$ DRP AAV-GFP, 5 days post-infection (DPI) with $10^5$ TCID$_{50}$ RSV, and HDM sensitized cotton rats were compared. Bars represent the mean histologic scores from a single experiment and standard deviations are represented by the bracketed bars. The means were compared using One-way ANOVA, and the asterisk indicates a significantly higher histologic inflammatory score compared to all other groups ($p < 0.05$). 5DPI PBS (n = 4), 5DPI AAV-G (n = 3), 12DPI AAV-G (n = 3), 19DPI AAV-G (n = 9), 19DPI AAV-GFP (n = 6), 26DPI AAV-G (n = 3), 5DPI RSV (n = 9), HDM (n = 4).

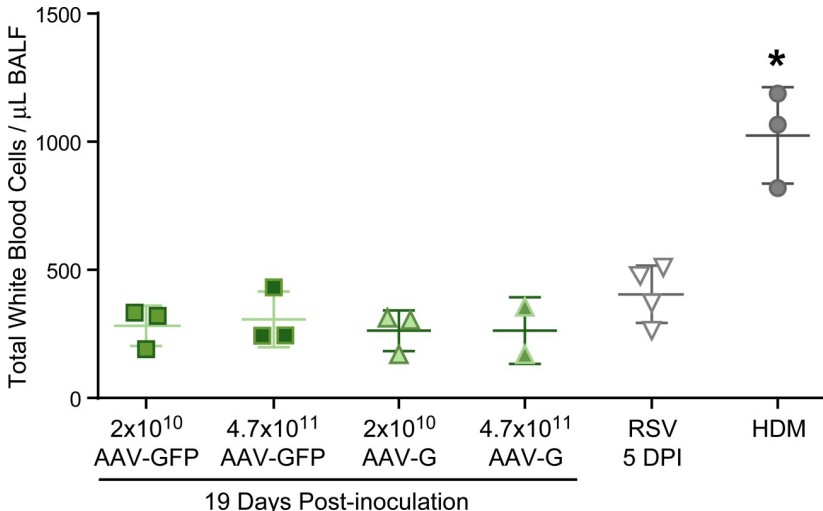

**Fig 5. Assessment of inflammation in cotton rats inoculated with AAV-G compared to AAV-GFP, RSV-infected and HDM-sensitized cotton rats via bronchoalveolar lavage.** The total white blood cells per μL of bronchoalveolar lavage fluid (BALF) were compared between cotton rats inoculated IN 19 days prior with $2 \times 10^{10}$ DRP AAV-GFP or AAV-G, or infected IN 5 days prior with $10^5$ TCID$_{50}$ RSV. Individual points represent the mean from one animal from a single experiment with the mean and standard deviations represented by the horizontal bracketed bars. The means were compared using One-way ANOVA, and the asterisk indicates significantly higher total white blood cell count within BALF compared to all other groups ($p < 0.05$). 19DPI AAV-GFP (n = 3), 19DPI AAV-G (n = 3), 5DPI RSV (n = 4), HDM (n = 3).

replication was assessed 4 days post-challenge in the nasal turbinates and lung tissues (Fig 6A and 6B). There was a significant decrease in viral replication (120-fold and 140-fold respectively) in the nose and lung of cotton rats immunized with AAV-G compared to those immunized with AAV-GFP; however, protection was not as high as in cotton rats previously infected with RSV. Only non-neutralizing RSV-G-specific antibodies were generated which peaked on day 19 (Fig 6C and 6D), the day of RSV challenge. However, though these antibodies did not neutralize RSV infection in the HEp2 cell line, it is possible that they could in others such as human airway epithelium (HAE). When the amount of total anti-RSV IgG was compared to post-challenge viral replication in the lung (Fig 6E), a significant positive correlation between antibody level and protection (Linear regression, $R^2 = 0.7321$, p = 0.0016) was found.

It was shown previously that CD8 T cells (in contrast to CD4 T cells) help to clear RSV during acute infection from cotton rat lungs [33,58–60].To determine whether CD8 T cells contributed to the protection observed after AAV-G immunization CD8 T cells were depleted with a monoclonal antibody specific for cotton rat CD8. In CD8 T cell depleted animals, AAV-G immunization did not protect cotton rats (Fig 6A and 6B). Therefore, CD8 T cells appear to play a significant role in the partial protection induced by AAV-G. In addition, total anti-RSV IgG level rather than neutralizing antibodies in the HEp2 *in vitro* system induced by immunization with AAV-G contributed to partial protection against RSV.

## The receptor-binding site is essential for antibody production and protection

It has been debated whether potential G vaccine candidates should include the receptor binding site (CX3C motif) of the G protein which could induce antibodies targeting or potentially stimulating an inflammatory response and should therefore be disrupted [5,6]. Similarly, concerns have been raised that the secreted form of the G protein in a candidate vaccine could potentially mimic the CX3CR1 chemokine and be pro-inflammatory [31,61,62]. Therefore, we sought to determine the effects of the receptor binding site and various forms of the G protein on the immune response and protection against RSV challenge. We compared cotton rats inoculated with different AAV-G vectors, one containing a mutated receptor-binding site G-C186S, one expressing the membrane bound G protein (mG), and one expressing the secreted G protein (sG) (Fig 2). Total anti-RSV IgG antibody levels were compared between cotton rats immunized intranasally with $2x10^{10}$ DRP AAV-GFP, AAV-G, AAV-G-C186S, AAV-mG, and AAV-sG, or infected with $10^5$ TCID$_{50}$ RSV (Fig 7A). In contrast to cotton rats immunized with AAV-G, cotton rats immunized with AAV-G-C186S, AAV-mG, or AAV-sG did not generate a G protein specific antibody response (p<0.001) and none of the forms of the G protein induced neutralizing antibodies (Fig 7B). In contrast, the G construct with an intact receptor binding site and expression of both the membrane bound and secreted form induced an antibody response. The protection after immunization with the various forms of the G protein were assessed by quantifying viral replication 4 days post-RSV challenge. Cotton rats immunized with AAV-G-C186S, which has the mutated receptor-binding site or AAV-sG were not protected against RSV challenge (Fig 7C and 7D). There was significant protection in the lung of cotton rats immunized with AAV-mG, but not to the level as those immunized with AAV-G.

## Addition of type I interferon increases antibody production and protection

As both the AAV vector and RSV G protein did not induce inflammation which might aid a G protein specific immune response, we investigated whether the addition of an immune

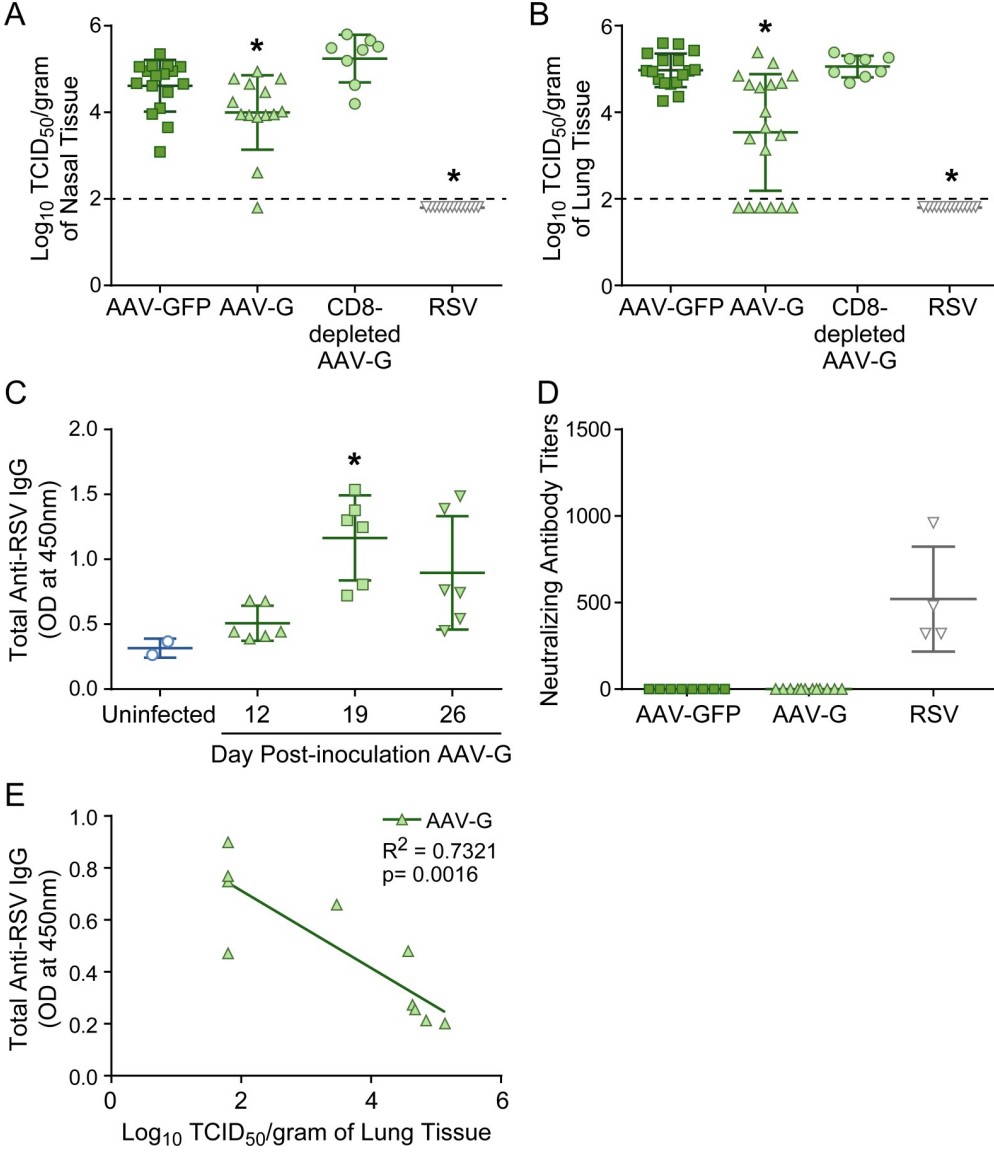

**Fig 6. Effect of immunization with RSV G protein on antibody production and protection against RSV challenge.**
(A-B) Four days post-challenge viral replication in the nose (A) and the lung (B). Post RSV challenge (IN $10^5$ $TCID_{50}$)
viral replication as quantified by a $TCID_{50}$ assay are compared between cotton rats immunized with AAV-GFP (IN
$2x10^{10}$ DRP), AAV-G (IN $2x10^{10}$ DRP), and RSV (IN $10^5$ $TCID_{50}$) 19 days prior. The CD8 depleted AAV-G group
represents the cotton rats repeatedly inoculated intraperitoneal with anti-cotton rat CD8 antibodies before and after
challenge. Individual points represent the mean $Log_{10}$ $TCID_{50}$/gram of tissue from one animal with the mean and
standard deviations represented by the horizontal bracketed bars from one experiment except AAV-GFP which was
data combined from 5 separate experiments, AAV-G which was combined data from 6 separate experiments, and RSV
which was data combined from 4 separate experiments; AAV-GFP (n = 17), AAV-G (n = 14–19), CD8 depleted
AAV-G (n = 8), RSV (n = 13). Asterisks indicate a significant difference from all other groups (p<0.05 by One-way
ANOVA), and the solid line across the graph represents the limit of detection for the assay. (C) Total anti-RSV IgG
levels as quantified using an indirect ELISA for sera from uninfected cotton rats and animals immunized with AAV-G
($2x10^{10}$ DRP) for 12, 19 and 26 days. Individual points represent the mean optical densities at 450nm from sera diluted
to 1:100 from one animal, with the mean and standard deviations represented by the horizontal bracketed bars;
uninfected (n = 2), 12DPI (n = 6), 19DPI (n = 6), 26DPI (n = 6). Asterisks indicate a significant difference from the
uninfected cotton rats (p<0.05 by One-way ANOVA). (D) Neutralizing antibody titers 19 days after immunization
with AAV-GFP, AAV-G or RSV. Individual points represent the mean neutralizing antibody titers from one animal
with mean and standard deviations represented by the horizontal bracketed bars; AAV-GFP (n = 8), AAV-G (n = 12),
RSV (n = 4). Data points represent the mean neutralization antibody titer from one animal, with data combined for
each group from 2–3 separate experiments. Cotton rats infected with RSV had significantly higher neutralizing
antibody titers; One-way ANOVA; p<0.0001. Whereas groups immunized with AAV-GFP and AAV-G did not have

detectable neutralizing antibodies. (E) Antibody production versus protection. The mean optical density at 450nm of total anti-RSV IgG levels were compared to against the mean $Log_{10}$ $TCID_{50}$/gram of lung tissue for each single animal immunized with AAV-G across 3 separate experiments, which is represent by an individual plot (n = 10). The line graph represents the best fit line as calculated by linear regression with the best fit, $R^2$ value (0.7321) and the p value, p = 0.0016 indicated on the graph. There is a significant correlation between high antibody levels and protection in AAV-G immunized animals.

stimulating substance such as IFNα would improve immunogenicity and protective capacity of the RSV-G protein by using an AAV expressing cotton rat IFNα.

Total anti-RSV IgG antibody levels from cotton rats co-immunized intranasally with AAV-G and AAV-INFα were compared to those immunized separately with AAV-INFα, AAV-G, AAV-GFP, or infected with RSV (Fig 8A). After co-immunization with AAV-G and AAV-INFα total RSV-specific antibodies reached the same level as after infection with RSV. However, these antibodies were not neutralizing (Fig 8B). Co-immunization with AAV-G and AAV-INFα did not lead to protection in the nose (Fig 8C) but led to a 10-fold lower viral load in the lung compared to those immunized with AAV-G alone (One-way ANOVA, p<0.0001) (Fig 8D). To determine if the addition of AAV-INFα affected the CD8 T cell response and contributed to the increased protection against challenge, we depleted the CD8 T cells via repeated intraperitoneal injections of anti-CD8 T cell antibodies before and after challenge in co-immunized animals. Depletion of CD8 T cells did not lead to abrogation of protection. These results indicate that the increase in protection that occurred after co-immunization with AAV-G and AAV-INFα was due to an increase in total anti-RSV antibody levels.

## Discussion

In this study we investigated the inflammatory and immunogenic properties of the G protein outside of the context of RSV. We developed the AAV vector system as a method for expressing foreign proteins in the cotton rat lung, a system which can be used for future application for localized transgenesis of the respiratory tract [63]. A characteristic of AAV is the production of low levels of the protein of interest for a prolonged period of time without the induction of inflammation [64,65]. Here, we have defined the optimal AAV serotype, AAV-5, for expression in cotton rat airway epithelium and demonstrated protein expression for at least four weeks.

Our studies demonstrated that the G protein itself does not induce an inflammatory response in the cotton rat lung. We investigated the *in vivo* pro-inflammatory effects of CX3CL-1 in comparison to the purified G protein ectodomain and found no significant difference in inflammation compared to cotton rats inoculated with RSV, CX3CL1 or PBS. However, studies with CX3CL-1 have used either *in vitro* assays or inflamed epithelial tissue to demonstrate the chemoattractant properties of CX3CL-1. Whereas we used an *in vivo* system in which CX3CL1 was throughout the cotton rat airway spaces, which could account for the lack of statistical significance in the increase in WBC within BALF in this group and signifies that this chemokine at physiologic levels is not the only player in the proinflammatory response against RSV within the lung. We therefore addressed the question of whether G protein has pro-inflammatory properties by expressing it in the natural RSV target cells. When the G protein was expressed at low levels for an extended period of time by epithelial cells, we still did not observe a significant inflammatory response in comparison to those inoculated with RSV or AAV-GFP. This is in contrast to previous studies that suggested that the G protein is a chemoattractant causing increased pulmonary inflammatory infiltrates due to its structural similarity to CX3CL1 [27–31,50,66–68]. However, prior studies did not distinguish inflammation associated with viral replication due to the G protein's necessary receptor-binding function versus the hypothesized chemotactic properties particularly attributed to the secreted

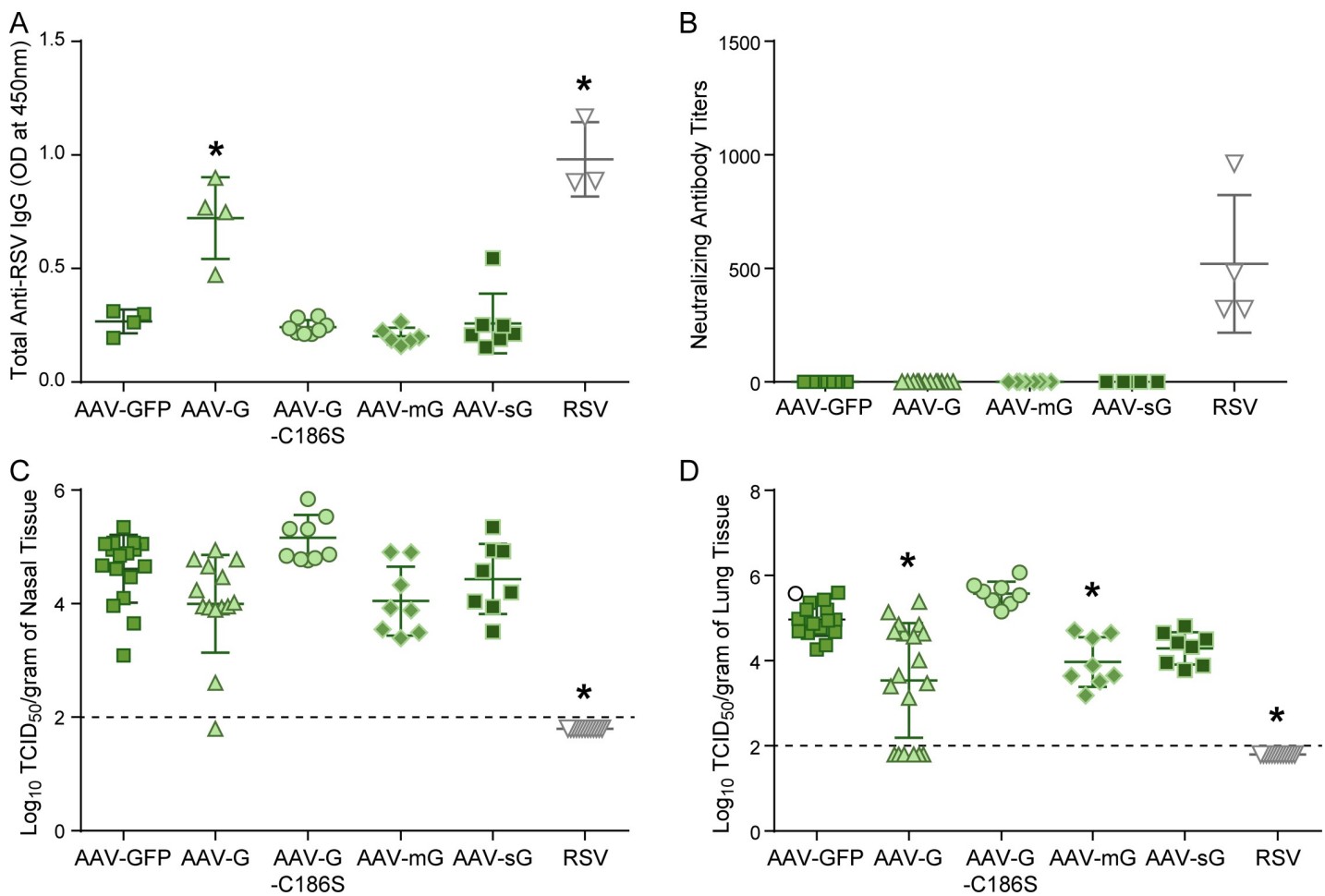

**Fig 7. The effect of various G protein forms on antibody production and protection.** (A) Total anti-RSV IgG levels as quantified by ELISA from 1:100 diluted sera collected 19 days post-inoculation in cotton rats immunized with $2 \times 10^{10}$ DRP IN AAV-GFP, AAV-G, AAV-G-C186S, AAV-mG, and AAV-sG or infected IN with $10^5$ TCID$_{50}$ RSV. Individual points represent the mean optical densities at 450nm from one animal with the mean and standard deviations represented by the horizontal bracketed bars, data is combined from 1–2 separate experiments; AAV-GFP (n = 4), AAV-G (n = 4), AAV-G-C186S (n = 8), AAV-mG (n = 6), AAV-sG (n = 7), RSV (n = 3). Asterisks indicate a statistical difference to all other groups (p<0.05, One-way ANOVA). (B) Neutralizing antibody titers after immunization with AAV-GFP, AAV-G, AAV-mG, AAV-sG, or RSV 19 days prior. Individual points represent the mean neutralizing antibody titers from one animal with the mean and standard deviations represented by the horizontal bracketed bars, data is combined from 1–3 separate experiments; AAV-GFP (n = 8), AAV-G (n = 12), AAV-mG (n = 8), AAV-sG (n = 4), RSV (n = 4). Cotton rats infected with RSV had significantly higher neutralizing antibody titers; One-way ANOVA, p< 0.0001. All other groups did not have detectable neutralizing antibodies. (C-D) Post-challenge viral replication in the nose (C) and the lung (D). Four days post RSV challenge (IN $10^5$ TCID$_{50}$) viral replication was quantified by a TCID$_{50}$ assay and compared between cotton rats immunized IN with $2 \times 10^{10}$ DRP AAV-GFP, AAV-G, AAV-G-C186S, AAV-mG, AAV-sG or infected IN with $10^5$ TCID$_{50}$ RSV 19 days prior. Individual points represent the mean log TCID$_{50}$/gram of tissue from one animal with the mean and standard deviations represented by the horizontal bracketed bars, data is combined from 1–6 separate experiments; AAV-GFP (n = 17), AAV-G (n = 14–19), AAV-G-C186S (n = 8), AAV-mG (n = 8), AAV-sG (n = 8), RSV (n = 13). Asterisks indicate a significant difference from AAV-GFP inoculated cotton rats (p<0.05 by One-way ANOVA), and the solid line across the graph represents the limit of detection for the assay.

form of the G protein [27–30,66–70]. Thus, based on our findings, the inflammatory response observed after RSV infection is most likely due to virus replication, other viral proteins rather than the chemoattractant properties of the G protein, or the host cell response to infection. Though the RSV G protein alone does not have a role in inflammation, it is possible that it still has some role in the context of RSV infection, especially as the receptor binding protein allowing for cell entry.

Many viral vector systems and live attenuated RSV vaccines typically express the G protein short-term and at higher levels than AAV resulting in the induction of neutralizing antibodies.

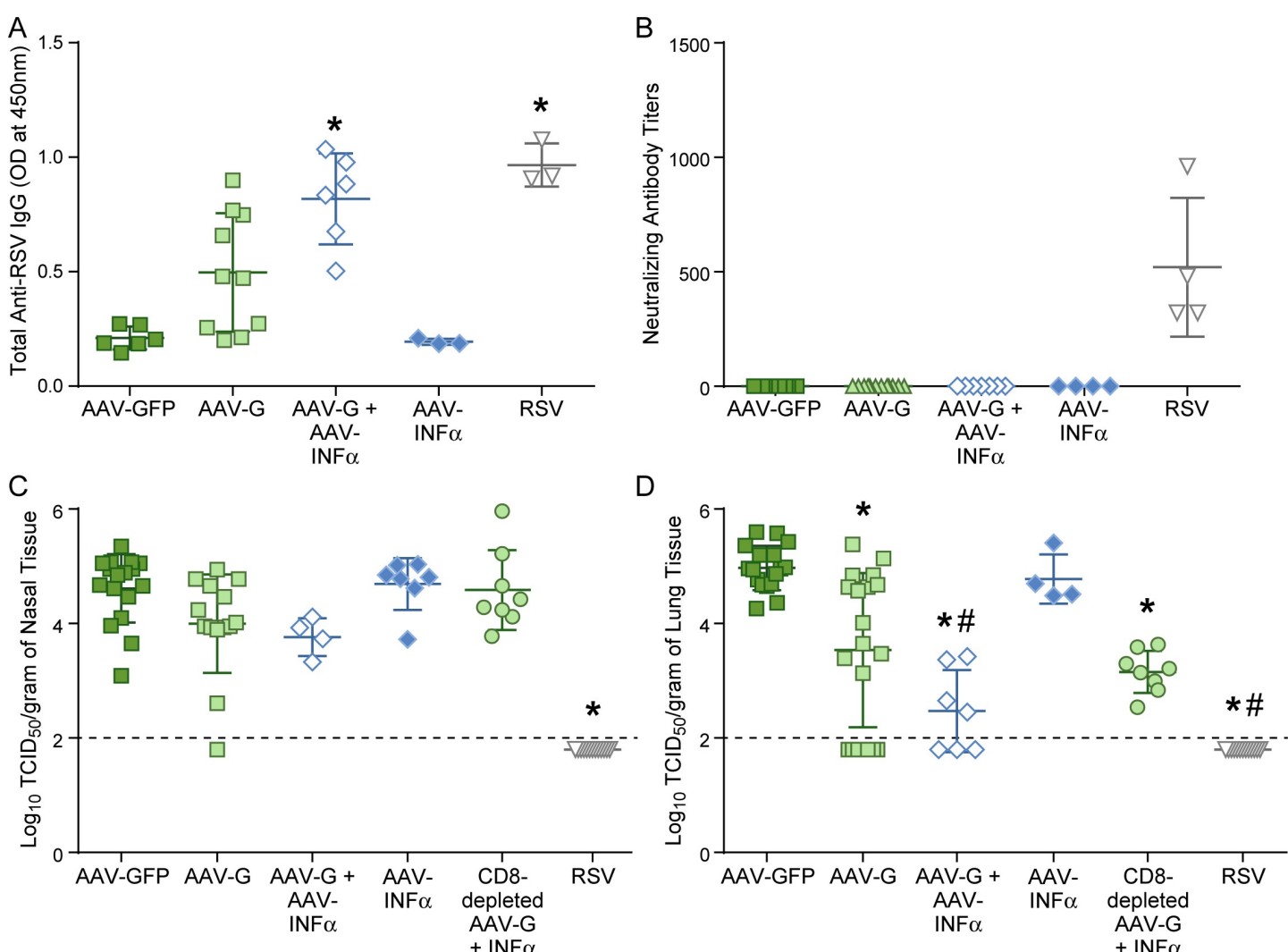

**Fig 8. The effects of type I interferon adjuvant on the adaptive immune response and protection against RSV challenge.** (A) Total anti-RSV IgG levels as quantified using an indirect ELISA from 1:100 diluted sera collected 19 days post-inoculation in cotton rats immunized with $2x10^{10}$ DRP IN AAV-GFP, AAV-G, AAV-INF$\alpha$, $2x10^{10}$ AAV-G and $2x10^{10}$ AAV-INF$\alpha$, or infected IN with $10^5$ TCID$_{50}$ RSV. Individual points represent the mean optical densities at 450nm from one animal with mean and standard deviations represented by the horizontal bracketed bars, data is combined from 1–2 separate experiments; AAV-GFP (n = 6), AAV-G (n = 10), AAV-INF$\alpha$ (n = 3), AAV-G + AAV-INF$\alpha$ (n = 6), RSV (n = 3). Asterisks indicate statistical significance to other groups without asterisks; One-way ANOVA, p<0.05. (B) Neutralizing antibody titers after immunization with AAV-GFP, AAV-G, AAV-G and AAV-INF$\alpha$, AAV-INF$\alpha$, or RSV 19 days prior. Individual points represent the mean neutralizing antibody titers from one animal with mean and standard deviations represented by the horizontal bracketed bars, data is combined from 1–2 separate experiments; AAV-GFP (n = 8), AAV-G (n = 12), AAV-INF$\alpha$ (n = 4), AAV-G + AAV-INF$\alpha$ (n = 7), RSV (n = 4). Cotton rats infected with RSV had significantly higher neutralizing antibody titers; One-way ANOVA, p< 0.0001. All other groups did not have detectable neutralizing antibodies. (C-D) Post-challenge viral replication in the nose (C) and the lung (D). Four days post RSV challenge (IN $10^5$ TCID$_{50}$) viral replication as quantified by a TCID50 assay are compared between cotton rats immunized AAV-GFP, AAV-G, AAV-G + AAV-INF$\alpha$, AAV-INF$\alpha$, or RSV 19 days prior. The CD8 depleted AAV-G and AAV-INF$\alpha$ group represents the cotton rats repeatedly inoculated intraperitoneal with anti-cotton rat CD8 antibodies before and after challenge. Individual points represent the mean log TCID$_{50}$/gram of tissue from one animal with mean and standard deviations represented by the horizontal bracketed bars, data is combined from 1–6 separate experiments; AAV-GFP (n = 17), AAV-G (n = 14–19), AAV-INF$\alpha$ (n = 4), AAV-G + AAV-INF$\alpha$ (n = 4–7), RSV (n = 13). Asterisks indicate a significant difference from AAV-GFP inoculated cotton rats (p<0.05 by One-way ANOVA), pound signs indicate a significant difference from AAV-G inoculated cotton rats (p<0.05 by One-way ANOVA), and the solid line across the graph represents the limit of detection for the assay.

However, the AAV vector system is unique in that it expresses low levels of protein for an extended period of time and did not induce a neutralizing antibody response against the G protein as detected in the HEp2 *in vitro* system. Thus, with this system we can address the question of whether in the absence of neutralizing antibodies there are other immune

mediators, such as CD8 T cells and non-neutralizing IgG antibodies that significantly contribute to protection against RSV infection. Our studies suggest that CD8 T cells are not only important for clearance [33,58–60], but also for protection against challenge with RSV.

Many studies attribute the G protein's ability to induce protection to its ability to induce neutralizing antibodies [14,16,28]. However, our data suggest that non-neutralizing anti-G antibodies as detected in a HEp2 *in vitro* system may play a significant role in protecting against RSV infection. Our findings that total anti-G antibody level correlate with protection is similar to findings in influenza virus infection. Infected individuals generate antibodies specific for the stalk of influenza hemagglutinin which are not virus-neutralizing but confer protection in a mouse model [71]. Similarly, other studies in cotton rats and mice have demonstrated that an anti-G IgG antibody responses associated with protection from RSV challenge with little to no detectable neutralizing antibodies [17,72]. It is likely that these antibodies contribute to protection from RSV by forming immune complexes that mediate ADCC. It is also likely that these protective non-neutralizing antibodies are against sites other than the central conserved region similar to influenza [72]. Therefore, the antibodies induced by inoculation with AAV-G may activate the complement system or antibody dependent cellular cytotoxicity (ADCC) as mechanisms for protection in addition to or rather than neutralization, and this should be examined in future studies using HAE cell lines. The G protein's immunogenicity was also found to be dependent on the receptor-binding site. Our study found that the receptor-binding site, specifically the CX3C motif that comprises the cysteine noose, is required for robust antibody production and protection. This is in line with other studies which also have identified this region as important for antibody production and protection [6,14,16,27,28]. The most likely explanation for the importance of the CX3C motif is that the structural integrity of the site must be maintained. Recently, it has been demonstrated that maintaining the adjacent domains and the regional structure of the G protein is important for antibody production [73], too. Therefore, though protective non-neutralizing antibodies may target regions other than the central conserved domain, altering the CX3C motif could change the conformation of the G protein and thus make these epitopes unrecognizable to those antibodies.

Expression of solely the secreted G protein did not induce an antibody response or protection, but co-expression of the secreted G along with the membrane bound form induced the greatest total antibody response. Lastly, due to the lack of an inflammatory response induced by the G protein, the addition of interferon alpha was found to increase antibody production and therefore protection.

Based on this study, we found no in vivo evidence for a pro-inflammatory function of the RSV-G protein. Immunization with a G protein with an intact receptor-binding site expressing both the membrane bound and the secreted form of the G protein conferred protection by inducing non-neutralizing antibodies as well as CD8 T cell responses specific for the G protein.

## Supporting information

**S1 Fig. Sequence of codon optimized RSV G protein with highlighted mutations.** Single bolded letters (C) represent nucleotides that were mutated for codon optimization. A series of three bolded letters represent key sites within the G protein gene that were mutated to produce various G protein constructs to be inserted into the AAV vector.
(TIF)

**S2 Fig. Pulmonary inflammation 48 hours post-inoculation or infection.** The total number of white blood cells per µL of bronchoalveolar lavage fluid (BALF) were compared at 12 hours,

24 hours, and 48 hours post-inoculation with 100μL of PBS, 50μg/mL purified G protein, 50μg/mL mouse CX3CL1, or $10^5$ TCID$_{50}$ RSV. Bars and brackets represent the mean and standard deviations. There was no significant difference between groups; Two-way ANOVA; $p >$ 0.05. PBS (n = 3–4), G protein (n = 4), CX3CL1 (n = 4), RSV (n = 2–4).
(TIF)

**S3 Fig. Bronchoalveolar lavage determined pulmonary inflammation with higher CX3CL1 inoculum.** The total number of white blood cells per μL of bronchoalveolar lavage fluid (BALF) were compared 48 hours post-inoculation with 100μL of PBS and 200μg/mL mouse CX3CL1 (n = 5). Bars and brackets represent the mean and standard deviations. There was no significant difference between groups; Student unpaired t test; $p >$ 0.05.
(TIF)

**S4 Fig. Immunohistochemical scoring of G protein expression after inoculation with AAV-G.** The semi-quantitative scoring of G protein expression via immunohistochemistry (IHC) with an anti-RSV antiserum in lung sections of cotton rats were compared. The mean and standard deviations are represented (n = 42-45/group). The asterisk indicates a significantly higher IHC score compared to all other groups (One-way ANOVA, $p < 0.05$).
(TIF)

**S1 Table. Semi-quantitative immunohistochemistry scoring system.**
(DOCX)

**S2 Table. Semi-quantitative histologic inflammatory scoring system.**
(DOCX)

**S3 Table. GFP expression in cotton rat tracheal cells after inoculation of different AAV serotypes.** Five days after inoculation of cotton rats with $10^{10}$ DRP of AAV serotypes 1, 5, and 6 expressing GFP, tracheal cells were isolated and tested for GFP expression. GFP expression was evaluated against the autofluorescence background of tracheal cells from animals inoculated with PBS. Groups of three animals were used.
(DOCX)

## Acknowledgments

The authors would like to thank Dr. La Perle and the Comparative Pathology and Mouse Phenotyping Shared Resource at The Ohio State University for the help with making and scanning the histology slides for this project. The authors would also like to thank Tim Vojt for his help with formatting the images for publication. Lastly, the authors would like to thank Nationwide Children's Hospital Viral Vector Core for producing the AAV vectors.

## Author Contributions

**Conceptualization:** Stefan Niewiesk.

**Data curation:** Margaret E. Martinez.

**Formal analysis:** Margaret E. Martinez.

**Funding acquisition:** Margaret E. Martinez, Stefan Niewiesk.

**Investigation:** Margaret E. Martinez, Devra Huey, Douglas McCarty, Stefan Niewiesk.

**Methodology:** Margaret E. Martinez, Douglas McCarty, Stefan Niewiesk.

**Project administration:** Stefan Niewiesk.

**Resources:** Cristina Capella Gonzalez, Mark E. Peeples, Douglas McCarty, Stefan Niewiesk.

**Supervision:** Stefan Niewiesk.

**Validation:** Margaret E. Martinez, Devra Huey.

**Visualization:** Margaret E. Martinez.

**Writing – original draft:** Margaret E. Martinez.

**Writing – review & editing:** Cristina Capella Gonzalez, Mark E. Peeples, Douglas McCarty, Stefan Niewiesk.

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
