## [Decision Letter · Decision Letter 0]

30 Sep 2020

PONE-D-20-27444

Immunogenicity and inflammatory properties of respiratory syncytial virus attachment G protein in cotton rats

PLOS ONE

Dear Dr. Martinez,

Thank you for submitting your manuscript to PLOS ONE. After careful consideration, we feel that it has merit but does not fully meet PLOS ONE’s publication criteria as it currently stands. Therefore, we invite you to submit a revised version of the manuscript that addresses the points raised during the review process.

We look forward to receiving your revised manuscript.

Kind regards,

Ralph A. Tripp

Academic Editor

PLOS ONE

Journal Requirements:

'M.E.M. was funded through the Genetech Fellowship under grant number GRT00044407 from Genentech (https://urldefense.com/v3/__https://www.gene.com/__;!!KGKeukY!kK079YjJDBxIYb4cCd4SEhIt14tL5fghqOewU8tfuKWiRtOTljnrsH7ujR1LFKsMRVtl$ ). D.H. and S.N. were funded through grant number P01AI112524 from the National Institute of Health National Institute of Allergy and Infectious Diseases (https://urldefense.com/v3/__https://www.niaid.nih.gov/__;!!KGKeukY!kK079YjJDBxIYb4cCd4SEhIt14tL5fghqOewU8tfuKWiRtOTljnrsH7ujR1LFHaJx-S8$ ). M.P and C.C-P. were funded through grant number P01 AI112524 and R01 AI093848 rom the National Institute of Health National Institute of Allergy and Infectious Diseases (https://urldefense.com/v3/__https://www.niaid.nih.gov/__;!!KGKeukY!kK079YjJDBxIYb4cCd4SEhIt14tL5fghqOewU8tfuKWiRtOTljnrsH7ujR1LFHaJx-S8$ ).'

We note that you received funding from a commercial source: Genetech

We also note that one or more of the authors are employed by a commercial company: Pfizer

b. Please also declare this commercial affiliation along with any other relevant declarations relating to employment, consultancy, patents, products in development, or marketed products, etc. in your updated Competing Interests Statement. 

Within your Competing Interests Statement, please confirm that this commercial affiliation does not alter your adherence to all PLOS ONE policies on sharing data and materials by including the following statement: "This does not alter our adherence to  PLOS ONE policies on sharing data and materials.” (as detailed online in our guide for authors http://journals.plos.org/plosone/s/competing-interests) . If this adherence statement is not accurate and  there are restrictions on sharing of data and/or materials, please state these.

Please note that we cannot proceed with consideration of your article until this information has been declared.

Additional Editor Comments:

Please address revisions suggested by the reviewers, particularly that anti-G antibodies previously described as “non-neutralizing” are actually neutralizing in a relevant infection model (HAEs).

Reviewers' comments:

Reviewer's Responses to Questions

**Comments to the Author**

1. Is the manuscript technically sound, and do the data support the conclusions?

Reviewer #1: Partly

Reviewer #2: Partly

2. Has the statistical analysis been performed appropriately and rigorously? 

Reviewer #1: Yes

Reviewer #2: Yes

3. Have the authors made all data underlying the findings in their manuscript fully available?

Reviewer #1: Yes

Reviewer #2: Yes

4. Is the manuscript presented in an intelligible fashion and written in standard English?

Reviewer #1: Yes

Reviewer #2: Yes

5. Review Comments to the Author

Reviewer #1: Summary:

This study explored the possibility that the RSV G protein alone induces pulmonary inflammatory responses. This study is motivated by the observation that antibodies against RSV G, or the deletion of the G gene from RSV, reduces inflammation during infection. The authors tested both purified RSV G protein and RSV G expressed by AAV in a cotton rat model and found no increase in pulmonary infection. Challenge with RSV showed decreased viral load in the lungs of animals that had previously received AAV-G or AAV-mG. There was significant correlation between high anti-G antibody levels as well as CD8+ Tcells and reduction of RSV load. Finally, mutation of the RSV G CX3C motif to CX3S eliminated its protective effects, suggesting that the structural integrity of RSV G is important for its protective effects.

Major comments:

1. The finding that RSV G alone does not induce inflammation does not necessarily mean that their data does “not support a role of the RSV G protein in inflammation” (line 462). The authors conclude from their data that anti-G antibodies block inflammation merely by reducing virus infection (lines 661-663), however it is still entirely possible that RSV G has a role in inflammation in the context of RSV infection. It is recommended to include this possibility in the discussion (lines 661-663) and also change line 462 to “…suggests that RSV G protein alone does not induce inflammation”

2. Throughout the manuscript, the authors describe the elicited anti-G antibodies as “non-neutralizing” because they do not inhibit RSV infection in HEp-2 cells. While this is a standard cell line for RSV growth, it is not necessarily an ideal model for RSV neutralization, especially for neutralizing anti-G antibodies. See for example https://pubmed.ncbi.nlm.nih.gov/26658574/ which shows that anti-G antibodies previously described as “non-neutralizing” are actually neutralizing in a relevant infection model (HAEs). It is recommended that the authors rewrite lines 489-490 “Only non-neutralizing RSV-G-specific antibodies were generated” to describe the caveat that the antibodies do not neutralize in this cell line, but it is possible that they neutralize in other cell types. To note, their own data supports that these anti-G antibodies are neutralizing (in vivo). We also recommend toning down the “non-neutralizing” statements throughout the manuscript (line 489, 532, 667, 678, 709, etc.)

Minor comments:

1. Studies with CX3CL1 use only the CX3CL1 chemokine domain, not the full ectodomain of CX3CL1, which could have different properties. This should at least be stated in the manuscript as a caveat to their studies.

2. Revise or eliminate lines 655-657 “This is in contrast to previous studies that suggested that the G protein is a chemoattractant causing increased pulmonary inflammatory infiltrates…” Similar to major comment #1 above, these studies do not necessarily contrast with previous studies. It is still possible that G protein is a chemoattractant causing increased pulmonary inflammatory infiltrates in the context of RSV infection.

Reviewer #2: This manuscript explores the role of the RSV G protein in lung inflammation, induction of total G protein antibodies, and protection from RSV challenge. The authors appropriately use cotton rats as an animal model and they express the G protein using an AAV vector. The manuscript shows data that challenges the view held by some in the field that it is the G protein that is responsible for lung inflammation after RSV infection. They found no evidence for induction of inflammation upon G expression in the animal lungs upon AAV-G intranasal inoculation. This conclusion is perhaps less convincing since their positive control, CX3CL1, also did not induce inflammation. They did however have a positive control, dust mite antigen.

To characterize antibody responses to the G protein, they used wild type G and mutant G proteins that expressed only the membrane bound form, the soluble form, or one defective in the receptor binding domain. They characterized total anti-G IgG, neutralizing antibodies, and levels of RSV in lungs and nasal tissue upon RSV challenge of the animals.

Major points

1. Conclusions based on absence of neutralizing antibody titers are puzzling. The authors used HEp2 cells in their neutralization assays. Cells that actually have adequate levels of CX3CR1, the G protein receptor, can be used to determine if anti-G antibodies can block virus entry (neutralize). Can the authors prove that HEp2 cells have adequate levels of the G protein receptor? Would the same results be found if primary human epithelial cells were used?

2. Comparisons of differential effects of wild type and mutant G proteins would be more convincing if the authors validated their constructions by showing proteins on gels, quantifying the relative levels of expression of the wild type and mutant proteins expressed from the AAV vector, and proving that the proteins were secreted or membrane associated or defective in CX3CR1 binding as predicted.

Minor Points

3. The authors should discuss why their positive control for inflammation, CX3CL1, did not cause inflammation.

4. Why did the methods include purifying G protein? This reagent was not mentioned in the results.

5. The numbers of animals/group used in some experiments were inadequate (3) or marginal (5). Cotton rats are not as inbred as mice thus larger groups are required for good statistics.

6. Lines 365-372 and Figure 2: discussion of the mutants should be in the same order as presented in the text.

7. Figure 3: how did the authors identify macrophages vs pneumocytes?

8. Line 455: sentence incomplete

9. In all figure legends, it is not clear if different points on graphs are the mean of different animals or the mean of multiple experiments on one animal. How many times were the determinations done?

10. Line 530: Do you mean figure 6, panels A and B?

11. Lines 701-703: this statement is contradictory. What is meant here?

6. PLOS authors have the option to publish the peer review history of their article (what does this mean?). If published, this will include your full peer review and any attached files.

Reviewer #1: No

Reviewer #2: No

---

## [Author Response · Author response to Decision Letter 0]

22 Jan 2021

Dear Editor, 

Thank you for the fast and thorough review of our manuscript. Overall, the reviewers had few critiques of our manuscript “Immunogenicity and inflammatory properties of respiratory syncytial virus attachment G protein in cotton rats”. Those critiques raised are addressed below. 

We have reformatted the manuscript as requested, and added additional information and edits requested by the reviewers’ to the reformatted version.

As requested we provide the following amended Competing Interests Statement

Genentech provided the corresponding author with a research fellowship that aided in the following research to be conducted. One of the authors was employed by Pfizer partly during the completion of the present manuscript. This does not alter our adherence to PLOS ONE policies on sharing data and materials. 

Also as requested we provide the following amended Funding Statement 

Dr. Douglas McCarty developed and provided the adeno-associated virus vectors used for the experiments in the following manuscript, as well as edited the manuscript for publication. Though Dr. Douglas McCarty is now employed by Pfizer, he did not conduct studies nor contribute to the study plan while at Pfizer. Only editing of the manuscript was performed while he was employed by Pfizer. Therefore, Pfizer did not contribute funding towards study design, materials, data collection or the majority of the salary for Dr. Douglas McCarty for the research presented in the following manuscript. The funder provided support in the form of salaries for authors [DM], but did not have any additional role in the study design, data collection and analysis, decision to publish, or preparation of the manuscript. The specific roles of these authors are articulated in the ‘author contributions’ section. 

As requested we provide the following amended Author contributions

Dr. Douglas McCarty developed and provided the adeno-associated virus vectors used for the experiments in the following manuscript, as well as edited the manuscript for publication. 

We hope that the manuscript is acceptable in its current form.

Sincerely, 

Margaret Martinez

Responses to reviewers:

Reviewer #1: 

Major Points

1. Conclusions based on absence of neutralizing antibody titers are puzzling. The authors used HEp2 cells in their neutralization assays. Cells that actually have adequate levels of CX3CR1, the G protein receptor, can be used to determine if anti-G antibodies can block virus entry (neutralize). Can the authors prove that HEp2 cells have adequate levels of the G protein receptor? Would the same results be found if primary human epithelial cells were used?

The Hep2 cell line is commonly used to determine neutralization activity of antibodies for various animal models as well as for humans to determine efficacy of vaccine candidates which is why the well-established technique was used [1–5,5–8]. It has been demonstrated that neutralization assays using Hep2 cell line and HAE cells both predict neutralization activity of anti-G IgG antibodies [1]. The same study found that neutralization is dependent on antibody dependent cytotoxicity and complement which is what we hypothesize is the mechanism of protection from the antibodies produced using the AAV system expressing the G protein. Therefore, it is reasonable to predict that a neutralization assay using HAE would similarly not detect neutralizing antibodies. 

Hep2 cells do not express CX3CR1 but heparin sulfate as receptor molecule.[9] 

2. Comparisons of differential effects of wild type and mutant G proteins would be more convincing if the authors validated their constructions by showing proteins on gels, quantifying the relative levels of expression of the wild type and mutant proteins expressed from the AAV vector, and proving that the proteins were secreted or membrane associated or defective in CX3CR1 binding as predicted.

As the AAV vector system produces little protein of interest for an extended period of time it is difficult to measure the amount of G protein expressed in vivo within cotton rat lungs. Immunohistochemistry was utilized to demonstrate expression of non-mutated G protein; however, the same technique could not be utilized for a secreted protein or a mutated receptor site G protein as the epitope would be altered. In vitro cell lines could be inoculated with AAV and western blot could be performed; however, the comparison of the secreted versus membrane bound form and potential misfolded forms might be difficult and not informative. 

Minor Points

3. The authors should discuss why their positive control for inflammation, CX3CL1, did not cause inflammation.

A line has been added at line #672-676

4. Why did the methods include purifying G protein? This reagent was not mentioned in the results.

The purified G protein was inoculated into cotton rats, with inflammation (BAL WBC count levels) compared to CX3CL1 and PBS 2 days following inoculation (Fig 1 and S2 fig).

5. The numbers of animals/group used in some experiments were inadequate (3) or marginal (5). Cotton rats are not as inbred as mice thus larger groups are required for good statistics. 

The sample size chosen is adequate for statistical analysis. As experiments were repeated throughout the study, the number of animals is increased and the experiments themselves are being repeated. This is inline with recommendations made in. [10]

Our colony of cotton rats are inbred as our lab internally bred the animals and have been kept so for years. Also, other studies have used similar sample sizes for their experiments using cotton rats [8,11]. And though cotton rats are not as inbred as mice, they are predictive of similar immune response as children which is likely because they do have some genetic heterogeneity like the population, they are modeling. Therefore, we argue our sample size is appropriate. 

6. Lines 365-372 and Figure 2: discussion of the mutants should be in the same order as presented in the text.

The order of these lines (370-372) have been re-arranged to be in the same order as the figure. 

7. Figure 3: how did the authors identify macrophages vs pneumocytes?

As the author is a trained pathologist, distinguishing macrophages (vacuolated mononuclear round cells within alveolar spaces) vs pneumocytes (squamous epithelial cells lining the air-blood barrier) was possible via histology. 

8. Line 455: sentence incomplete

Lines 459-464 have been altered to be complete sentences. 

9. In all figure legends, it is not clear if different points on graphs are the mean of different animals or the mean of multiple experiments on one animal. How many times were the determinations done?

Data points represent the mean from one animal from one experiment for BAL and histology experiments, except when indicated specifically for the AAV immunogenicity and protection studies. Lines 353-354, 451, 477, 513-516, 529-531, 535-536, 584, 590, 599-560, 632-633, 638-639, and 650 had additions to indicate the number of experiments the data points were combined from for the various groups. 

10. Line 530: Do you mean figure 6, panels A and B?

Yes. This has been corrected. 

11. Lines 701-703: this statement is contradictory. What is meant here?

This has been changed (now line 733) to the following: Expression of solely the secreted G protein did not induce an antibody response or protection, but co-expression of the secreted G its expression along with the membrane bound form induced the greatest total antibody response.

Reviewer #2: 

Major comments:

1. The finding that RSV G alone does not induce inflammation does not necessarily mean that their data does “not support a role of the RSV G protein in inflammation” (line 462). The authors conclude from their data that anti-G antibodies block inflammation merely by reducing virus infection (lines 661-663), however it is still entirely possible that RSV G has a role in inflammation in the context of RSV infection. It is recommended to include this possibility in the discussion (lines 661-663) and also change line 462 to “…suggests that RSV G protein alone does not induce inflammation” 

These lines have been changed 469 and 688-691. 

2. Throughout the manuscript, the authors describe the elicited anti-G antibodies as “non-neutralizing” because they do not inhibit RSV infection in HEp-2 cells. While this is a standard cell line for RSV growth, it is not necessarily an ideal model for RSV neutralization, especially for neutralizing anti-G antibodies. See for example https://pubmed.ncbi.nlm.nih.gov/26658574/ which shows that anti-G antibodies previously described as “non-neutralizing” are actually neutralizing in a relevant infection model (HAEs). It is recommended that the authors rewrite lines 489-490 “Only non-neutralizing RSVG-specific antibodies were generated” to describe the caveat that the antibodies do not neutralize in this cell line, but it is possible that they neutralize in other cell types. To note, their own data supports that these anti-G antibodies are neutralizing (in vivo). We also recommend toning down the “nonneutralizing” statements throughout the manuscript (line 489, 532, 667, 678, 709, etc.)

These lines have been changed (692-720). 

Minor comments:

1. Studies with CX3CL1 use only the CX3CL1 chemokine domain, not the full ectodomain of CX3CL1, which could have different properties. This should at least be stated in the manuscript as a caveat to their studies.

Lines 333-336 have been added. 

2. Revise or eliminate lines 655-657 “This is in contrast to previous studies that suggested that the G protein is a chemoattractant causing increased pulmonary inflammatory infiltrates…” Similar to major comment #1 above, these studies do not necessarily contrast with previous studies. It is still possible that G protein is a chemoattractant causing increased pulmonary inflammatory infiltrates in the context of RSV infection.

The following statement (lines 682-685) describes how the G protein may cause inflammation in the context of RSV infection through its function as the receptor binding protein. Therefore, revising the statement “This is in contrast to previous studies that suggested that the G protein is a chemoattractant causing increased pulmonary inflammatory infiltrates…” is not necessary. 

1. Cortjens B, Yasuda E, Yu X, Wagner K, Claassen YB, Bakker AQ, et al. Broadly Reactive Anti-Respiratory Syncytial Virus G Antibodies from Exposed Individuals Effectively Inhibit Infection of Primary Airway Epithelial Cells. J Virol. 2017;91. doi:10.1128/JVI.02357-16

2. Lee H-J, Lee J-Y, Park M-H, Kim J-Y, Chang J. Monoclonal Antibody against G Glycoprotein Increases Respiratory Syncytial Virus Clearance In Vivo and Prevents Vaccine-Enhanced Diseases. PLoS One. 2017;12: e0169139. doi:10.1371/journal.pone.0169139

3. Lee Y-N, Suk Hwang H, Kim M-C, Lee Y-T, Cho M-K, Kwon Y-M, et al. Recombinant influenza virus carrying the conserved domain of respiratory syncytial virus (RSV) G protein confers protection against RSV without inflammatory disease. Virology. 2015;476: 217–225. doi:10.1016/j.virol.2014.12.004

4. Marcandalli J, Fiala B, Ols S, Perotti M, de van der Schueren W, Snijder J, et al. Induction of Potent Neutralizing Antibody Responses by a Designed Protein Nanoparticle Vaccine for Respiratory Syncytial Virus. Cell. 2019;176: 1420-1431.e17. doi:10.1016/j.cell.2019.01.046

5. Tang A, Chen Z, Cox KS, Su H-P, Callahan C, Fridman A, et al. A potent broadly neutralizing human RSV antibody targets conserved site IV of the fusion glycoprotein. Nat Commun. 2019;10: 4153. doi:10.1038/s41467-019-12137-1

6. Van Der Plas JL, Verdijk P, Van Brummelen EMJ, Jeeninga RE, Roestenberg M, Burggraaf J, et al. Prevalent levels of RSV serum neutralizing antibodies in healthy adults outside the RSV-season. Hum Vaccin Immunother. 2020;16: 1322–1326. doi:10.1080/21645515.2019.1688040

7. Verdijk P, van der Plas JL, van Brummelen EMJ, Jeeninga RE, de Haan CAM, Roestenberg M, et al. First-in-human administration of a live-attenuated RSV vaccine lacking the G-protein assessing safety, tolerability, shedding and immunogenicity: a randomized controlled trial. Vaccine. 2020;38: 6088–6095. doi:10.1016/j.vaccine.2020.07.029

8. Wang D, Phan S, DiStefano DJ, Citron MP, Callahan CL, Indrawati L, et al. A Single-Dose Recombinant Parainfluenza Virus 5-Vectored Vaccine Expressing Respiratory Syncytial Virus (RSV) F or G Protein Protected Cotton Rats and African Green Monkeys from RSV Challenge. J Virol. 2017;91. doi:10.1128/JVI.00066-17

9. Johnson SM, McNally BA, Ioannidis I, Flano E, Teng MN, Oomens AG, et al. Respiratory Syncytial Virus Uses CX3CR1 as a Receptor on Primary Human Airway Epithelial Cultures. PLoS Pathog. 2015;11: e1005318. doi:10.1371/journal.ppat.1005318

10. Vaux DL. Research methods: Know when your numbers are significant. Nature. 2012;492: 180–181. doi:10.1038/492180a

11. Widjojoatmodjo MN, Boes J, van Bers M, van Remmerden Y, Roholl PJM, Luytjes W. A highly attenuated recombinant human respiratory syncytial virus lacking the G protein induces long-lasting protection in cotton rats. Virol J. 2010;7: 114. doi:10.1186/1743-422X-7-114

---

## [Editor Report · Decision Letter 1]

26 Jan 2021

Immunogenicity and inflammatory properties of respiratory syncytial virus attachment G protein in cotton rats

PONE-D-20-27444R1

Dear Dr. Martinez,

We’re pleased to inform you that your manuscript has been judged scientifically suitable for publication and will be formally accepted for publication once it meets all outstanding technical requirements.

Kind regards,

Ralph A. Tripp

Academic Editor

PLOS ONE

Additional Editor Comments (optional):

The revised manuscript is acceptable.
---

## [Editor Report · Acceptance letter]

8 Feb 2021

PONE-D-20-27444R1 

Immunogenicity and inflammatory properties of respiratory syncytial virus attachment G protein in cotton rats 

Dear Dr. Martinez:

I'm pleased to inform you that your manuscript has been deemed suitable for publication in PLOS ONE. Congratulations! Your manuscript is now with our production department. 

Kind regards, 

on behalf of

Dr. Ralph A. Tripp 

Academic Editor

PLOS ONE